DISCOVERY REPORT

# *Streptomyces venezuelae* uses secreted chitinases and a designated ABC transporter to support the competitive saprophytic catabolism of chitin

Anne van der Meij[1], Hannah Tyrrell[1], Dustin J. Sokolowski[2], Evan M. F. Shepherdson[3], Marie A. Elliot[3], Justin R. Nodwell[1]*

1 Department of Biochemistry, University of Toronto, Toronto, Ontario, Canada, 2 Ontario Institute for Cancer Research, Toronto, Ontario, Canada, 3 Department of Biology, McMaster University, Hamilton, Ontario, Canada

* justin.nodwell@utoronto.ca

## Abstract

More than a billion tons of chitin are produced on earth each year. Chitin is rich in nitrogen and carbon, making it a valuable resource in competitive microbial ecosystems. However, almost all chitin is found in large, insoluble structures like insect and crustacean exoskeletons. For this material to enter a microorganism's primary metabolism, it must be degraded extracellularly through a saprophytic process. The extracellular nature of this process means that liberated oligomers may also become accessible to other microorganisms. How microbes navigate this challenge in terrestrial ecosystems remains largely unclear. Here, we show that *Streptomyces venezuelae* thrives on raw, insoluble chitin as its sole carbon and nitrogen source, outperforming glucose in metabolic activity and sporulation. This was facilitated by a chitinolytic system encompassing up to 10 chitinases and the DasABC chitobiose importer. While deleting some chitinases affected growth on chitin, others did not, implying some degree of functional redundancy. A *dasBC* null mutation conferred a severe growth defect suggesting that chitobiose is a key breakdown product during chitin-based metabolism in *S. venezuelae*. The DasABC transporter also played a crucial role in preventing the built-up of chitobiose extracellularly, thereby restricting its access to *Bacillus subtilis* in co-cultures. Given the global ubiquity of *Streptomyces* in soil, this pathway likely plays a significant role in soil ecology as well as carbon and nitrogen turnover on a global scale.

## Introduction

Saprophytes obtain nutrients by breaking down dead and decaying organic material [1]. This nutrient source contains polymers like (hemi)cellulose, starch, lignin, chitin, protein, fatty acids, DNA, and RNA [2,3]. Some of these polymers aggregate to form

**Data availability statement:** RNA sequencing data have been deposited in ArrayExpress under accession E-MTAB-15268. All other data supporting the findings of this study are included within the manuscript and its Supporting information files.

**Funding:** This work was supported by grant #019.212EN.013 to AvdM from the Dutch Research Counsel, and by grant #RGPIN-2021-03861 to JRN from the Natural Science and Engineering Research Counsel of Canada. The funders had no role in study design, data collection and analysis, decision to publish, or preparation of the manuscript.

**Competing interests:** I have read the journal's policy and the authors of this manuscript have the following competing interests: The authors disclose that a patent application related to chitobiose production by the dasBC deletion mutant has been planned.

**Abbreviations:** AUC, area under the curve; CFU, colony-forming unit; LB, lysogeny broth; LC–MS, liquid chromatography–mass spectrometry; MYM, maltose-yeast extract-malt extract.

large, insoluble solids that must be broken down to oligomers that can be imported across membranes to enter primary metabolism. Importantly, this polymer decomposition must happen outside the cell, necessitating the secretion of enzymes that are metabolically expensive to produce. These saprophytic processes have been studied extensively in fungi and bacteria [4–9] because of their ecological importance and industrial utility [10–12].

*Streptomyces* are filamentous, Gram-positive microorganisms. They are most often studied for their specialized metabolism, which is a rich source of antibiotics and other medicines [13–16]. Other subjects of biological interest include their exploratory motility [17], filamentous growth habit, sporulation capabilities [18,19], and increasingly, their mutualistic relationships with eukaryotes, including insects [20–24]. Importantly, streptomycetes inhabit soil everywhere on earth and it is thought that they play ecological roles in the decomposition of organic matter. They have been described as saprophytes and are known to secrete enzymes into their extracellular milieu, a hallmark of saprophytic digestion [25]. Indeed, genomic and culturing studies have demonstrated that streptomycetes possess a complex branched carbon metabolism, suggesting that they can consume a wide variety of organic materials [26,27].

A saprophytic lifestyle opens up vast stores of energy-rich material to microorganisms; however, it comes with unique risks. In nature, unlike in most laboratories, sources of bioavailable carbon and nitrogen are subject to fierce competition. It has been estimated that every gram of soil on Earth harbors thousands of distinct microbial species and billions of cells [28]. Such resource competition is a topic of recurring interest in microbial ecology, with a well-characterized example being competition for iron, which is insoluble in most soils. Microorganisms acquire iron by secreting siderophores that form tight complexes with the atom [29,30]. These iron-siderophore complexes are then imported, and the iron is liberated for metabolic use. These iron-binding probes are a valuable "public good" [31], and consequently, siderophore piracy is common; competing microbes can import siderophores (and their associated iron) that had been generated by their neighbors [29]. It has been suggested that "private" siderophores, which can only be imported by their producers, can be important in competitive environments [32]. Another form of resource competition involves "exploiters"—microbes that rely on polymer breakdown products generated by the enzymatic activity of other organisms rather than producing these breakdown products themselves [33]. As a consequence, the loss of valuable nutrients to competing microorganisms represents an inherent challenge of saprophytic growth.

Chitin is a naturally abundant polymer, consisting of chains of N-acetyl glucosamine linked by β-1-4 bonds. Notably, more than 1 billion tons of chitin are generated globally every year [34]. It is a primary constituent of insect and crustacean exoskeletons as well as fungal cell walls. Unlike the more common polymer cellulose, the chitin monomer includes both carbon and nitrogen, making it an especially valuable resource in nature. The most abundant polymer form, α-chitin, consists of aligned polymers having numerous polymer–polymer hydrogen bonds, and can be moulded into an extraordinary diversity of shapes [35]. Chitin is insoluble in water and is

resistant to hydrolysis and other spontaneous degradative pathways [36]. Thus, leveraging this resource for growth necessitates degradation via extracellular digestion.

It has been demonstrated that *Streptomyces* can degrade colloidal chitin—an acid-treated form of chitin powder that does not occur naturally [37,38]. Importantly, compared to untreated chitin powder, colloidal chitin is more soluble, has a larger surface area, and greatly improved bioavailability [39]. Due to these properties, colloidal chitin incorporated into agar media has been used since the early 1960s to culture and isolate *Streptomyces* from soil samples [40,41]. Nevertheless, studies also indicate that neither colloidal nor raw chitin enhanced actinomycete isolation more effectively than water agar alone [37,42,43], suggesting that chitin was not the main determinant of growth outcomes in these studies. Moreover, *Streptomyces* research on chitin has primarily concentrated on the chitinolytic activity of specific chitinases, usually using colloidal chitin as a chitinase inducer [44–49]. In these studies, media were typically enriched with additional nitrogen and/or carbon sources, making it difficult to assess *Streptomyces* capacity to metabolize this potential source of nutrients.

Our interest in the chitin degradative capabilities of *Streptomyces* was spurred by work showing that *Streptomyces* species generate odorants that can attract insects [21,50]. We found that a subset of these streptomycetes generate specialized metabolites that allow them to kill the attracted insects or their offspring [50]. We wondered whether this might represent a mechanism by which *Streptomyces* attracts potential food sources.

In this study, we show that insects, insect exoskeletons, and chitin isolated from shrimp exoskeletons, can serve as sole sources of carbon and nitrogen for efficient growth and sporulation by *Streptomyces venezuelae*. Other common soil bacteria such as *Bacillus subtilis, Pseudomonas aeruginosa* and *Rhodococcus jostii* are unable to use this material as their sole food source. Efficient growth on chitin is supported by multiple chitinases. Importantly, we find that the DasABC chitobiose transport system [51] plays two crucial roles in chitin-based growth. First, it imports key products of saprophytic chitin degradation, since mutants with an impaired transporter system are severely affected in growth. Additionally, we show that the activity of the DasABC transporter prevents *B. subtilis* from gaining access to chitin breakdown products during co-culture with *S. venezuelae,* effectively safe-guarding its chitobiose supply. The *Streptomyces* chitinolytic system is conserved in many species [27,52, this work] and we find that widely diverged species retain the capacity for growth on chitin as primary nutrient. This suggests that chitin degradation is likely to be a major factor in *Streptomyces'* capacity to compete in nature and that this metabolism is a significant component of the global carbon and nitrogen cycles.

## Materials and Methods

### Strains, media, and culture conditions

An overview of all strains, plasmids, and cosmids used can be found in Table A in S1 Text. *Streptomyces venezuelae* NRRL B-65442 was grown on solid maltose-yeast extract-malt extract (MYM) medium [53] at 30°C for 2 days for spore stock generation. Spore stocks for *Streptomyces coelicolor* M145, *Streptomyces avermitilis* K139 [54], WAC303 [55], and WAC288 [55] were similarly prepared with incubation times of up to 4 days. *Bacillus subtilis* (PY79) BDR2662 i::P*rpsJ*-*mCherry* was kindly provided by Prof. D.Z. Rudner and cultured in lysogeny broth (LB) unless stated otherwise. *Escherichia coli* DH5α, *Pseudomonas aeruginosa* PAO1, and *Rhodococcus jostii* RHA1 were cultured in LB unless stated otherwise.

Growth on whole grasshoppers was performed by adding 10 mL of autoclaved tap water to a single autoclaved grasshopper (Fresh Feeders—Exotic Nutrition) placed in a petri dish. Spores were added at a density of $10^6$ spores per mL, and petri dishes were incubated statically at 30 °C.

For liquid cultures, bacteria were typically inoculated at a density of $10^6$ spores/mL unless stated otherwise. The minimal chitin medium contained, per liter: 5 g of chitin flakes derived from shrimp (CAS: 1398-611-4), 0.6 g of $MgSO_4 \cdot 7H_2O$, 2 g of $(NH_4)_2SO_4$ (when specified), 150 mL of $NaH_2PO_4/K_2HPO_4$ buffer (0.1 M, pH 6.8), 1 mL of trace elements solution [53], and 850 mL of tap water. The minimal exoskeleton medium contained 5 g of crushed grasshopper exoskeleton (Fresh Feeders - Exotic Nutrition), which replaced the chitin in the minimal chitin medium. Glucose medium consisted of 5 g of

D-glucose and always included 2 g of $(NH_4)_2SO_4$ as a nitrogen source, replacing the chitin in the minimal chitin medium. Cultures were generally grown in 5 mL volumes within 25 mL glass tubes and incubated at 225 RPM and 30°C. The raw data for all measurements described herein are contained in S2 Data.

## Microscopy

Phase-contrast images were taken with a Zeiss Axio Lab 5 upright microscope, equipped with a Zeiss Axiocam 208 color camera. Morphological studies on *S. venezuelae* grown on grasshoppers by Scanning Electron Microscopy were performed using a prisma environmental scanning electron microscope by Thermo Scientific. Pieces of grasshopper with bacterial biomass were cut and fixed with 1.5% glutaraldehyde (1 h). Subsequently, samples were dehydrated (70% acetone for 15 min, 80% acetone for 15 min, 90% acetone for 15 min, and 100% acetone for 15 min), and critical point dried. Hereafter, the samples were coated with gold using a sputter coater before imaging.

## Resazurin assays

Resazurin assays were used to measure metabolic activity. Spores were inoculated in either chitin or glucose medium containing resazurin (CAS:62758-13-8) (15 µg/mL). One milliliter was transferred to a 48-well-plate and covered with a Breathe-Easy sealing membrane. The plate was then incubated at 30 °C with low orbital shaking in a Synergy plate reader for 12 h. Emission at 590 nm was measured every 10 min (excitation: 550 nm). The curves represent the mean of three independent 1 mL cultures for each time point.

## RNA sequencing

*S. venezuelae* was grown in triplicate in 20 mL minimal glucose medium and 20 mL minimal chitin medium, both supplemented with $(NH_4)2(SO_4)$ for 24 h. 5 mL of each culture was pelleted, frozen in liquid nitrogen, and kept at −80 °C before RNA extraction. RNA was extracted from the cell pellets using the Qiagen RNeasy Micro Kit (Qiagen, no. 74004) according to the manufacturer's protocol, with a minor modification: 0.5 mL of 0.5 mm glass beads were added to samples in lysis buffer followed by bead beating (six times 1 min beating versus one minute on ice). RNA integrity was verified by gel electrophoresis before being sent for sequencing at Genome Quebec. There, total RNA was quantified and its integrity was assessed using a LabChip GXII (PerkinElmer) instrument. rRNA was depleted from 250 ng of total RNA using QIAseq FastSelect (-5S/16S/23S Kit 96rxns). cDNA synthesis was achieved with the NEBNext RNA First Strand Synthesis and NEBNext Ultra Directional RNA Second Strand Synthesis Modules (New England BioLabs). Library preparation was then completed using the NEBNext Ultra II DNA Library Prep Kit for Illumina (New England BioLabs). Adapters and PCR primers were purchased from New England BioLabs. Libraries were quantified by Kapa Illumina GA with Revised Primers-SYBR Fast Universal kit (Kapa Biosystems). Average DNA fragment sizes were determined using a LabChip GXII (PerkinElmer) instrument. The libraries were normalized, pooled, and then denatured in 0.05 N NaOH and neutralized using hybridization buffer (HT1). The pool was loaded at 200 pM on an Illumina Nova-Seq S4 lane using the NovaSeq Xp Workflow as per the manufacturer's recommendations. The run was performed for 2 × 100 cycles (paired-end mode). A phiX library (Illumina) was used as a sequencing control and mixed with libraries at 1% level. Base calling was performed with RTA v3. Program bcl2fastq2 v2.20 was then used to demultiplex samples and generate fastq reads.

RNA sequencing reads were aligned to the *S.venezuelae* genome assembly (GCA_001886595.1_ASM188659v1) using the STAR aligner [56]. First, we indexed the genome assembly using "STAR genomeGenerate" with "—genomeSAindexNbases 8" to support bacterial genomes [56]. We then used "STAR alignReads" to align paired end reads with "--alignIntronMin 20 --alignIntronMax 19 --outFilterMultimapNmax 20" parameters which prevents STAR from incorporating introns and gap junctions into alignments, thereby making STAR compatible with bacterial genomes [56]. Quality control of raw aligned reads were summarized using fastqc [57]. We the then filtered PCR duplicates with "Picard MarkDuplicates"

using VALIDATION_STRINGENCY=LENIENT [58] and filtered reads with a quality score <30 with "samtools view" [59]. To view these reads on the Interactive Genome Viewer (IGV) [60], we counted the reads per kilobase per million (PRKM) normalized alignments and converted the alignment "bam" file to a "bigwig" file using DeepTools "bamCoverage" [61]. These filtered reads were then assigned to genes in the "GCA_001886595.1_ASM188659v1_genomic" genome annotation using featureCounts "-s 1 -Q 30" to allow for high-quality strand-specific alignments [62]. Filtered and counted RNA-seq reads were then subjected to data normalization. First, we filtered rRNA genes, which account for a high proportion of sequenced reads and can impact downstream analysis. Then, we RPKM normalized the remaining counts using the edgeR R package [63]. Genes with 5 or fewer reads in at least 2 of the 3 biological replicates were excluded. Gene-expression between samples was measured using Pearson's correlation and plotted into a heatmap with the Pheatmap R package (https://cran.r-project.org/web/packages/pheatmap/pheatmap.pdf) (Fig A in S1 Text). Then, we performed Principal Component Analysis (PCA) between samples and plotted the first two PCs with ggplot2 (Fig B in S1 Text). Differential expression analysis between glucose and chitin conditions were performed with edgeR [63], with gas the "control" condition. Differentially expressed genes (DEGs) were then plotted using a Manhattan plot using ggplot2. Raw sequencing data is included in S1 Data.

### Chitinase annotation and architecture

We initially identified six proteins as putative GH18 family chitinases (vnz_02735; vnz_05055; vnz_23100; vnz_24855; vnz26400; vnz35060) and one GH19 family chitinase (vnz_16685) by using the *Streptomyces* Annotation Server (https://strepdb.streptomyces.org.uk). We used the catalytic domains of these putative chitinases as queries for BLAST searches against the *S. venezuelae* genome and this led to the identification of three additional putative GH18 family chitinases (vnz_12680; vnz_23445; vnz_12765). To predict their enzyme architecture, we utilized UniProt's integrated databases (InterPro; PROSITE; Pfam; SUPfam) and identified putative binding domains, catalytic domains, and putative transmembrane regions (Table B in S1 Text), which were then structurally examined using the alpha fold plug-in.

### Phylogeny

We conducted phylogenetic analyses using Molecular Evolutionary Genetics Analysis 11 (MEGA11) software [64]. We performed an alignment of the amino acid sequences for the catalytic domains of the nine GH18 family chitinases present in the *S. venezuelae* genome, and these same chitinases together with 15 previously identified chitinases from the GH18 family from various bacteria [65,66], utilizing the MUSCLE plug-in with its default settings. We applied the maximum likelihood algorithm, coupled with the Jones-Taylor-Thornton substitution model, to deduce the unrooted phylogenetic trees. To evaluate the confidence in the branching patterns of the resulting trees, we performed a bootstrap analysis with 500 replicates. The chitinases from *S. venezuelae* were categorized into subfamilies based on their phylogenetic relationships with the previously characterized chitinases.

### Chitinase conservation

We conducted a protein BLAST (BLASTP) search using full-length protein sequences of all the identified *S. venezuelae* chitinases, recombinase (vnz_26845), and vnz_04420 (gene in chloramphenicol biosynthetic gene cluster) to assess conservation across bacterial taxa, with a particular focus on the *Streptomycetaceae* family. The BLAST search was performed against the NCBI protein database using stringent thresholds: query coverage of 80%–100%, sequence identity of 50%–100%, and an *E*-value of ≤1E−05. To ensure comprehensive retrieval of homologous sequences, the Maximum Target Sequences parameter was set to 5,000.

The total number of hits and the proportion of hits from the *Streptomycetaceae* family for each query sequence are summarized in Table E in S1 Text.

## Construction of *S. venezuelae* mutants

Gene deletions were generated using ReDirect technology [67]. In short, coding sequences on a cosmid vector carrying large fragments (30–40 kb) of *S. venezuelae* genomic DNA were replaced by an *oriT*-containing apramycin resistance cassette (PCR amplified from pIJ773) with primers as indicated in Table C in S1 Text. The mutant cosmids were introduced into the methylation-deficient *Escherichia coli* strain ET12567 carrying the conjugation helper plasmid pUZ8002, followed by conjugation into *S. venezuelae*. Exconjugants were screened for double-crossover events, and gene deletions were confirmed by PCR using various combinations of primers positioned to confirm the presence of the apramycin resistance cassette and absence of the wild-type gene (Table C in S1 Text).

Mutant strains were complemented by expressing the wild-type gene *in trans,* after cloning them into the integrating vector pMS82 [68] behind the constitutive *ermE** promoter to ensure high levels of transcription [69] using restriction enzymes NdeI (Promega) and XhoI (Promega) and T4 ligase (New England BioLabs). Primers are listed in Table C in S1 Text.

## Colony-forming unit (CFU) assays for *Streptomyces*

Spores were inoculated in 5 mL of minimal chitin medium or MYM liquid medium in 25 mL glass tubes to minimize shear stress and facilitate chitin attachment. Cultures were incubated at 30 °C with shaking at 225 rpm. After three days of incubation in MYM medium and six days in chitin medium, cultures were vortexed three times for 10 seconds each, and 10-fold serial dilutions were prepared. Two microlitre spots were inoculated to MYM agar and incubated overnight. CFU counts were determined from the highest dilution factor that supported growth. Bar graphs represent the averages of three independent cultures.

## Liquid chromatography–mass spectrometry (LC–MS)

To detect chitobiose in culture supernatants, spent medium from 4-day cultures of wild-type S*. venezuelae* and the *dasBC* deletion strain were filtered and dried. The dried spent medium was resuspended in half the original volume in LC–MS grade water. One-hundred microlitres was injected into an Acquity UPLC BEH C18 column (1.7 μm, 2.1 × 50 mm) in parallel with blank controls. The separation gradient was at 0.2 mL/min, 95% A for 1 min, increasing linearly to 60% A over 22 min, to 5% A at 24 min and held at 5% for 2 min: A—water + 0.1% formic acid and B—acetonitrile + 0.1% formic acid. Samples were analyzed by electrospray ionization with MS/MS on selected masses (447.16; 425.18; 244.08) on a QTOF (Waters Xevo G2S-QTOF).

## *S. venezuelae*—*B. subtilis* co-cultivation

*B. subtilis* BDR2662 *sacA*::P*rpsJ-mCherry* was kindly provided by Prof. D. Rudner. Bacteria were cultured overnight at 30 °C with shaking at 225 rpm. The overnight culture was then diluted 1:50 into MYM or minimal chitin medium with $(NH_4)2(SO_4)$. Similarly, $10^6$ spores/mL of either wild-type *S. venezuelae* or the *dasBC* deletion mutant were added to generate a co-culture. One milliliter of each condition was transferred to a 48-well plate and covered with a Breathe-Easy sealing membrane. The plate was incubated at 30 °C with low orbital shaking in a Synergy plate reader for 16–48 h, measuring fluorescence emission at 620 nm every 10 min (excitation: 550 nm). The curves represent the mean of three 1 mL cultures for each time point. To determine the CFUs of *B. subtilis* in co-cultures, a 1:1000 dilution of an overnight *B. subtilis* culture was added to 1 mL of MYM or chitin medium containing $(NH_4)_2(SO_4)$ with $10^6$ spores/mL of either wild-type *S. venezuelae* or the *dasBC* deletion mutant in a 48-well plate. Cultures were incubated as static cultures for 3 days, followed by preparation of 10-fold serial dilutions. A 5 μL sample from each dilution was spotted onto LB agar and incubated overnight. CFU counts derived by determining the highest dilution factor that supported growth. Bar graphs represent the averages of three independent cultures.

## Assessment of *S. venezuelae* spent medium toxicity on *B. subtilis*

*B. subtilis* BDR2662 *sacA*::P*rpsJ-mCherry* (PY79) was cultured overnight at 37 °C with shaking at 225 rpm. The overnight culture was then diluted 1:1000 into LB. To every 450 microliters of LB and *B. subtilis* cells, 50 µL of 10× concentrated *S. venezuelae* spent medium was added (see below). For each culture, 0.5 mL were transferred to a 48-well plate and covered with a Breathe-Easy sealing membrane. The plate was incubated at 37 °C with high double-orbital shaking in a Synergy plate reader for 16–48 h, measuring fluorescence emission at 590 nm every 10 min (excitation: 550 nm). The curves represent the mean of three 0.5 mL cultures for each time point. Concentrated spent media samples were prepared from 3 mL of spent medium of *S. venezuelae* wild-type and *dasBC* deletion strains grown on chitin for 16 h, which were filtered and evaporated. The resulting residue was dissolved in 300 µL milliQ water. Spent media concentrates were prepared in triplicate.

## Results

### *Streptomyces* can grow using chitin as sole source of carbon and nitrogen

We carried out two experiments to determine whether *Streptomyces venezuelae* could grow using raw chitin as carbon and nitrogen source. First, we applied spores to sterilized grasshopper carcasses. We observed vigorous growth, with the insect being covered by a dense hyphal mat within 2–3 days (Fig 1A). We imaged this growth using electron microscopy and observed a dense mycelium with individual hyphae appearing to grow along the length of the exoskeleton (Fig C in S1 Text). We measured the grasshopper masses before and after 6 weeks of incubation with *S. venezuelae* spores, normalizing to an uninoculated control insect. The combined mass of the remaining material and the *Streptomyces* hyphal mat showed a 27% decrease relative to the insect prior to incubation (Fig 1B), which could be due to consumption and $CO_2$ release during respiration by *S. venezuelae*.

The grasshopper exoskeleton is composed primarily of chitin in addition to proteins and other carbohydrates [70], which are all potential sources of carbon and nitrogen to *Streptomyces*. To investigate whether the exoskeleton alone would support bacterial growth we dried the grasshoppers extensively and dissected away the intra-abdominal material to liberate exoskeleton. We then crushed the dissected exoskeleton, sterilized the resulting flaky powder, and added this to sterile phosphate buffer (pH 6.8) and trace elements, resulting in an exoskeleton medium. Again, *S. venezuelae* was able to grow efficiently, illustrated by the formation of aggregates consisting of exoskeletal fragments and tightly-associated hyphae within 2 days (Fig 1C). Together, these results show that *Streptomyces* can grow using dead insect exoskeleton as sole nutrient, supporting their status as saprophytes.

Second, we created a defined medium consisting of commercially sourced, practical grade α-chitin flakes from shrimp, suspended in phosphate buffer and trace elements. Colloidal chitin as well as additional carbon sources, such as agar, amino acids, or yeast extract, were deliberately excluded [40,41,44–49,71]. Again, we found that *S. venezuelae* could grow efficiently on this defined medium (Fig 1D) and completed its life cycle with the production of $10^8$ spores per mL after 5–6 days (Fig 1E and inset 1G). Moreover, we found that in inoculated static chitin cultures, over 50% of the weight had escaped the system after 6 weeks of incubation, again likely due to $CO_2$ release during respiration (Fig 1F). Note that these measurements include the *Streptomyces* biomass, suggesting that a greater proportion of chitin had been metabolized.

To determine if this trait is specific to *S. venezuelae*, we tested four other *Streptomyces* species for growth in this restrictive medium: the evolutionarily diverged *Streptomyces coelicolor* and *Streptomyces avermitilis,* alongside the environmental isolates WAC288, and WAC303 from the Wright Actinomycetes collection [55]. All four species generated substantial biomass after 5 days (Fig D in S1 Text). In contrast, *E. coli*, *Pseudomonas aeruginosa*, *Bacillus subtilis*, and *Rhodococcus jostii*, all of which are common soil inhabitants, were unable to grow on this medium (Fig E in S1 Text). This indicates that the growth observed for the diverse *Streptomyces* species is not a universal property of bacteria, including soil dwellers.

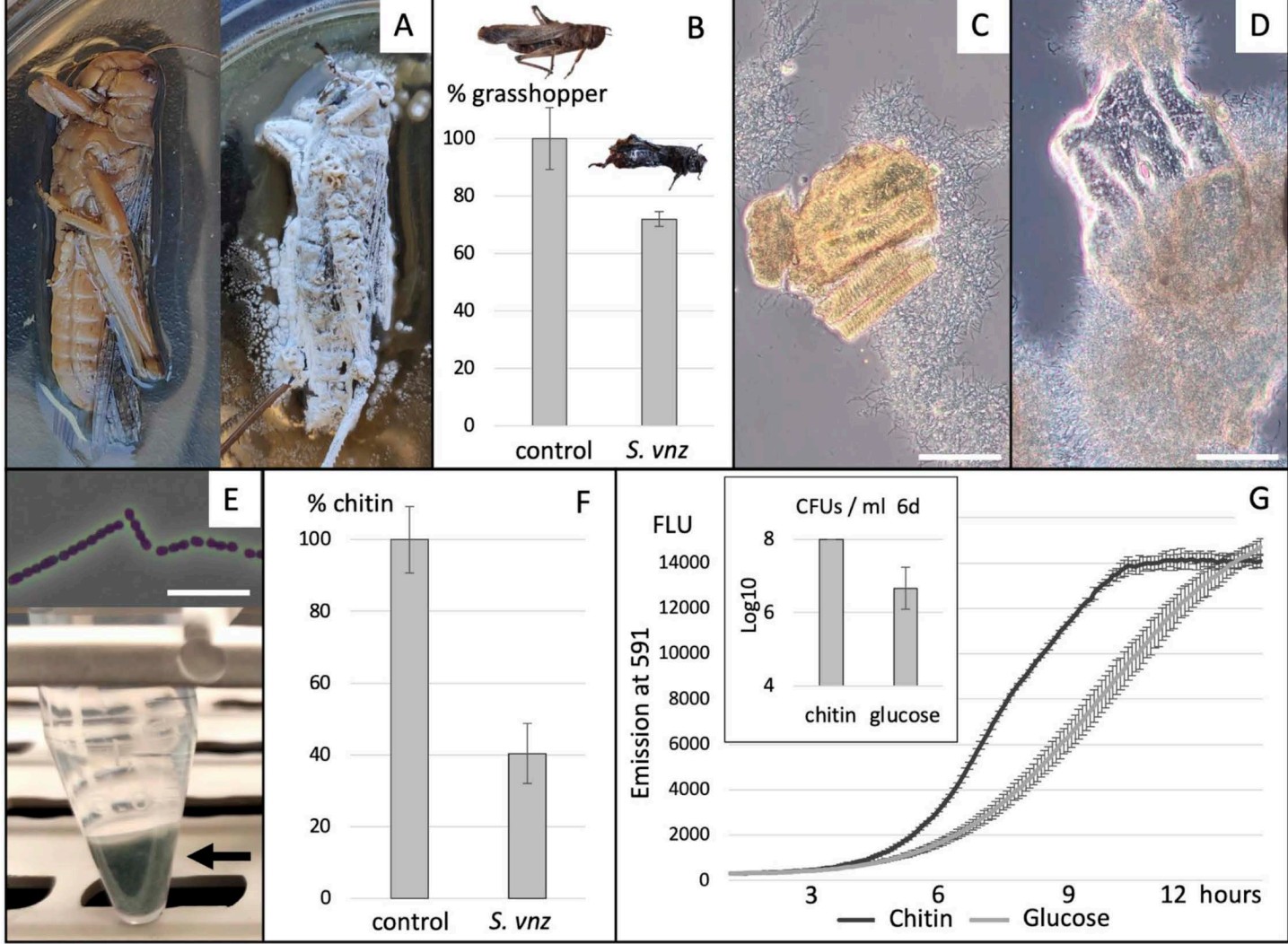

**Fig 1. *Streptomyces venezuelae* growth on insect (-exoskeleton) and chitin. A)** *S. venezuelae* overgrowing a grasshopper, visualized by an extensive hyphal mat covering the exoskeleton of the insect. **B)** Growth of *S. venezuelae* (*S. vnz*) lead to a 27% loss of grasshopper biomass after an incubation period of 6 weeks. Bars represent the averages of eight biological replicates and the error bars show standard deviations. **C)** *S. venezuelae* hyphal mass coating a piece of insect exoskeleton **D)** *S. venezuelae* growing on a piece of chitin. **E)** Growth on chitin supports sporulation by *S. venezuelae*, illustrated by the green spore pellet (bottom panel—arrow) and the observation of spores by microscopy (top panel). **F)** Growth of *S. venezuelae* (*S. vnz*) on chitin results in biomass loss of over 50% after an incubation period of six weeks. Bars represent the averages of five biological replicates and the error bars show standard deviations. **G)** The metabolic marker resazurin reaches its peak in fluorescence earlier for cultures grown on chitin compared with glucose. Insert: bar graph showing CFUs per milliliter after 6 days of growth on either chitin medium or glucose medium. Scalebars: C and D: 50 microns. E: 5 microns. The data underlying this figure can be found in S2 Data.

Optical density was not a feasible means of quantifying *S. venezuelae* growth under these conditions due both to the filamentous nature of *Streptomyces* cells, and the presence of light scattering chitin flakes. We therefore used resazurin, which is taken up by metabolically active cells and reduced by cellular respiratory dehydrogenases to resorufin, which fluoresces at 590 nm [72]. In this way, we compared the metabolic activity of *S. venezuelae* in chitin medium with cultures using glucose and ammonium as carbon and nitrogen sources (see materials and methods). We found that the cells reached their peak reducing potential in the chitin medium 25% faster than in the glucose medium (Fig 1G). Furthermore,

growth on the defined glucose medium did not support efficient spore formation. This is supported by CFU measurements after six days of growth: chitin-grown cultures yield approximately $10^8$ spores per milliliter, whereas glucose-grown cultures reach only $10^6$ to $10^7$ CFUs per milliliter (Fig 1G, inset).

Collectively, these data show that *S. venezuelae* readily grows and completes its life cycle using raw chitin as primary source of carbon and nitrogen. These observations suggest that *Streptomyces* have not only incredibly specialized metabolic capabilities, but have also evolved a remarkable primary metabolism, that enables them to take advantage of this abundant but insoluble nutrient source.

## Growth on chitin results in a distinct gene expression profile

To determine how growth on chitin influences gene expression, we carried out an RNA sequencing experiment, comparing growth in liquid minimal chitin medium to growth in liquid minimal glucose medium. In this instance, we added ammonium to both growth conditions to focus primarily on the use of chitin (versus glucose) as a source of carbon. Cultures were harvested after 24 h of growth to allow for the accumulation of sufficient amount of biomass for both growth conditions. RNA was extracted, followed by the preparation of cDNA libraries, which were then sequenced.

The transcript profiles for growth on glucose and chitin were significantly different, with 2,681 out of 7,462 genes exhibiting changes of 2-fold or greater (FDR ≤ 0.05) (Fig 2A and S1 Data). Out of those, 1,460 genes had significantly higher transcript levels during growth on chitin.

The annotations of the 40 genes with the lowest false discovery rates and the highest comparative transcript levels during growth on chitin are listed in Table D in S1 Text and shown as groups of orthologues genes (COGs) (Fig 2B). We found that the majority of genes could either be attributed to groups C, G, and S, which correspond to "Energy production and conversion" (C), "Carbohydrate metabolism and transport" (G), and "Function unknown" (S), respectively. As we previously observed stronger reducing power in cells grown on chitin (Fig 1G), we examined the genes within the "energy production and conversion" group more closely. We found that this group primarily comprised genes encoding subunits of the NADH-oxidoreductase complex (complex I) (Table D in S1 Text), a component of the respiratory chain in both bacteria and eukaryotes. In *S. venezuelae,* complex I comprises 14 core subunits (*vnz_21055–vnz_21130*), which is standard in bacteria. In addition, the *S. venezuelae* genome encodes 9 subunit homologs (*vnz_21265–vnz_21310*), which is uncommon for bacteria [73]. Notably, the transcript levels of the homologs were unchanged between chitin and glucose, while those of the 14 core subunits were significantly higher during growth on chitin (Table D in S1 Text). This suggested that metabolic energy production is altered in bacteria grown on chitin.

## Classification and functional specialization of *S. venezuelae* chitinases

We next investigated the genes in group G for "carbohydrate metabolism and transport", aiming to identify enzymes involved in chitin catabolism. Notably, this group, consisting of 11 genes, was predominantly composed of chitinase-annotated genes, which accounted for 7 of them (Table D in S1 Text). We queried the whole *S. venezuelae* genome and identified at least 10 putative chitinases in the *S. venezuelae* genome (see materials and methods). Of these, one had been previously investigated for its antifungal activity [74]; however, aside from that, none of them has been characterized.

We analyzed the protein architecture, phylogeny, conservation, and expression patterns of all putative *S. venezuelae* chitinases. Most bacterial chitinases have a catalytic domain belonging to glycoside hydrolase family 18 (GH18), which hydrolyzes the β-1,4 linkages between the N-acetylglucosamine (GlcNAc) residues. We found that nine *S. venezuelae* chitinases belong to the GH18 family, while one belongs to the glycoside hydrolase family 19 (GH19), which is primarily associated with plant chitinases but had previously been reported in *Streptomyces* too [75]. Eight of these chitinases (vnz_02735; vnz_05055; vnz_12680; vnz_16685; vnz_23445; vnz_24855; vnz_26400; vnz_35060) have signal peptides indicating that they are secreted proteins, while the remaining two chitinases (vnz_12765 and vnz_23100) contain hydrophobic residues at the N-terminus of the protein, suggesting that they are possibly membrane-anchored. The seven

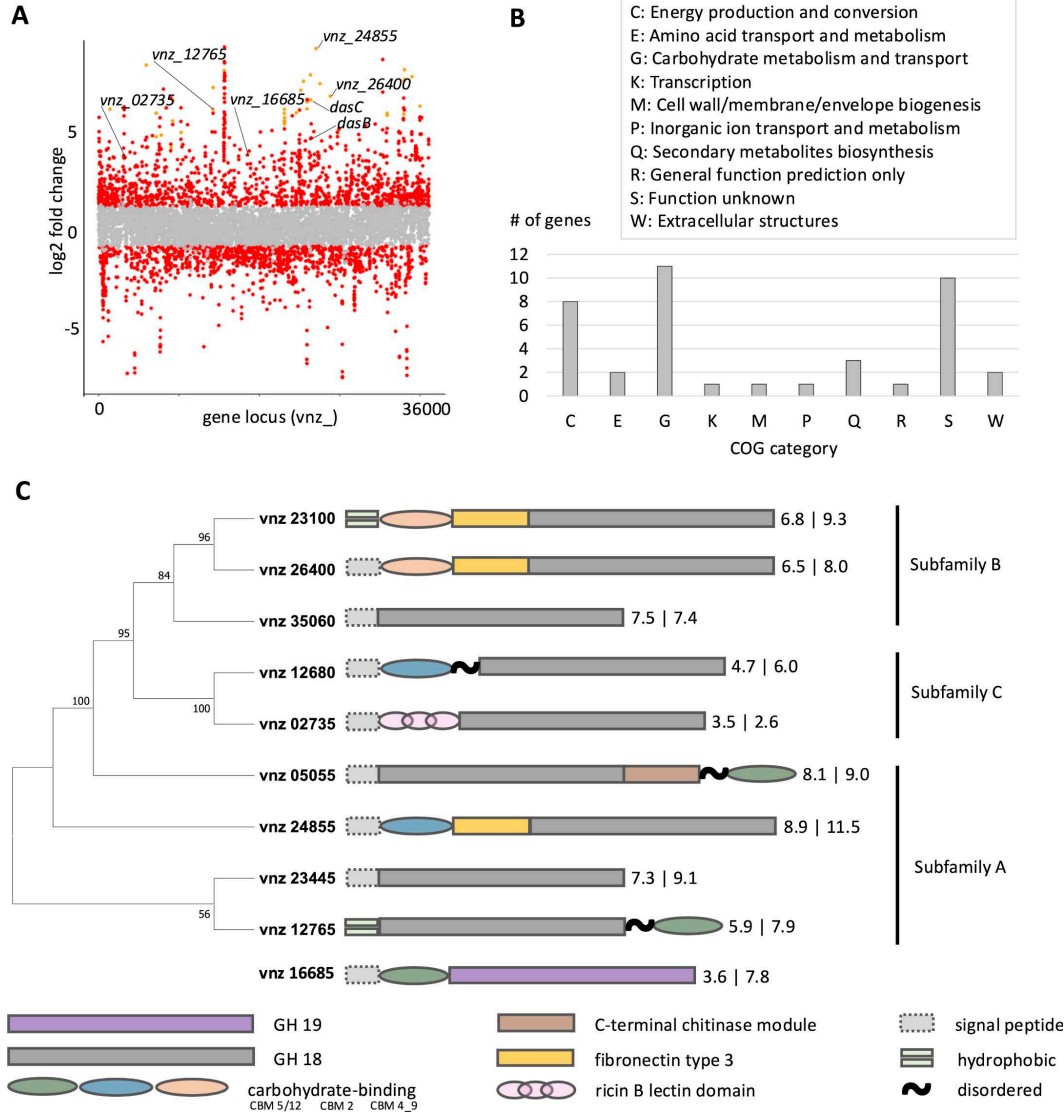

**Fig 2. Transcriptional response of *Streptomyces venezuelae* during growth on chitin. A)** Manhattan plot of *S. venezuelae* on glucose vs. chitin after 24 h of growth. Each dot represents a gene. Red dots are significantly differentially expressed with an FDR cutoff of 0.05. Genes with a positive Log2 fold change are upregulated during growth on chitin. Orange dots represent the 40 genes that are most significantly upregulated (lowest False Discovery Rate) during growth on chitin (Table D in S1 Text). Genes that were further investigated in this study are labeled. **B)** The 40 genes that are most significantly upregulated during growth on chitin (orange dots) have been assigned a Cluster of Orthologues Genes (COG) group. **C)** Phylogeny, domain architecture, and expression of *S. venezuelae* chitinases. The genome encodes one GH19 and nine GH18 chitinases with various chitin-binding domains. Subfamily assignments are shown. The first number indicates log$_2$ fold change (chitin vs. glucose); the second shows log$_2$ counts per million. The data underlying this Figure can be found in S1 and S2 Data. Please open S3 Data to view the tree-file.

chitinase hits from the RNA sequencing data (vnz_05055; vnz_12765; vnz_23100; vnz_23445; vnz_24855; vnz_26400; vnz_35060) are all members of the GH18 family.

GH18 family chitinases can be further divided into subfamilies A, B, and C [65,76], on the basis of their function and evolution: members of GH18 subfamily A are usually processive chitinases that release GlcNAc or chitobiose (a dimer of GlcNAc) from the chitin polymer. In contrast, subfamily B enzymes are primarily endo-acting chitinases that cleave the

polymer into oligosaccharides. Finally, subfamily C chitinases have a distinct evolutionary lineage. By constructing an unrooted phylogenetic tree of known bacterial GH18 chitinases from subfamilies A, B and C together with *S. venezuelae* GH18 chitinases (Fig F in S1 Text), we showed that vnz_05055; vnz_12765; vnz_23445; vnz_24855 belong to group A, vnz_23100; vnz_26400 and vnz_35060 to group B, and vnz_02735 and vnz_12680 to group C (Fig 2C). Individual chitinases were distinguished by their distinct carbohydrate binding domains, which included various families of carbohydrate-binding modules, fibronectin type 3 modules, and, in the case of vnz_02735, a ricin B lectin domain, which is unusual for chitinases. While it is not uncommon for *Streptomyces* species to have 10 or more chitinases encoded in their genomes [27,52]; the specific biological roles of these enzymes in the bacteria remain largely unexplored. Indeed, protein BLAST analysis demonstrated that the *S. venezuelae* chitinases are common within the *Streptomycetaceae* family. All *S. venezuelae* chitinases yielded between 1,700 and 4,600 hits within the *Streptomycetaceae* family under stringent conditions (query coverage 80%–100%, sequence identity 50%–100%, and an *E*-value threshold of ≤1E−05), reflecting their high conservation (Table E in S1 Text and materials and methods). These numbers approach the level of the highly conserved recombinase RecA, which produced 5,600 hits. In contrast, a gene from the chloramphenicol biosynthetic gene cluster, found only in a subset of *Streptomycetaceae*, resulted in approximately 100 hits, showing limited distribution within the family. These data highlight the widespread presence of multiple chitinases in members of the *Streptomyces* genus.

### Enzymes for chitin degradation and product import

To address the importance of the putative chitinase genes to grow on chitin, we constructed null mutations in genes encoding two members from group A (vnz_24855 and vnz_12765), one group B (vnz_26400), one group C (vnz_02375), and the one GH19 enzyme (vnz_16685), and compared the growth of the mutant strains to their wild-type parent on defined chitin medium and MYM medium, which contains maltose, yeast extract, and malt extract as carbon and nitrogen sources [53].

The null mutants of subfamily A chitinases vnz_24855 (which had the highest differential transcripts abundance on chitin amongst all chitinases) and vnz_12765 had impaired growth in chitin medium as measured by CFUs assays after 6 days (Fig 3A). As a result of efficient sporulation, the wild type produced an average of $10^8$ CFU/mL in chitin culture at this time point, whereas the *vnz_24855* and *vnz_12765* null mutants yielded approximately $10^6$ CFU/mL, as sporulation in these strains had only just begun. It is important to note that the mutants are capable of reaching higher CFU levels, but require additional time to do so. Both mutant phenotypes could be complemented by introducing a functional copy of their respective genes into their associated null mutant *in trans* (Fig 3A). Importantly, the growth of both mutants was similar to wild-type in MYM liquid cultures (Fig H in S1 Text), supporting a specific role for these two chitinases during growth on chitin. These data, together with the RNA sequencing data, suggests that the subfamily A chitinases in *S. venezuelae* are important for growth on chitin flakes when provided as only carbon and nitrogen source.

We then assessed the role of group B chitinases in supporting growth on chitin. A null mutation in vnz_26400, the group B chitinase with the highest differential transcript levels in the RNA sequencing data, had similar growth defects as observed for the group A mutants. The vnz_26400 deletion mutant yielded ~$10^6$ CFUs per mL compared to $10^8$ for the wild-type when grown on chitin (Fig 3A), but grew like wild-type in MYM cultures (Fig H in S1 Text). The null mutant strains for those chitinases that showed less abundant transcript levels (subfamily C: vnz_02735 and GH19: vnz_16685) did not exhibit the same CFU drops when grown on chitin (Fig 3A).

In order to grow on the liberated oligosaccharides resulting from the action of extracellular chitinases, *S. venezuelae* must have a mechanism to import these metabolites. We noted that transcripts from the *dasABCD* operon—encoding the ABC transporter for chitobiose uptake in *S. coelicolor* [51] - were some of the most highly differentially expressed genes when comparing growth on chitin and glucose medium (Table D in S1 Text). To test their importance to growth on chitin, we deleted the *dasB* and *dasC* genes, which encode the transmembrane subunits of the transporter, and characterized the growth of the resulting strain as we did for the chitinase mutants.

PLOS Biology

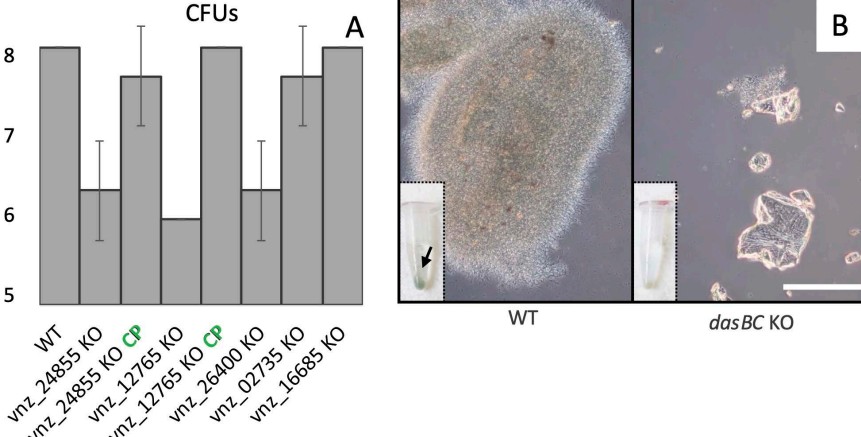

**Fig 3. Colony-forming units (CFUs) of wild-type and mutant strains grown on chitin. A)** Bar graph: Wild-type (WT) *Streptomyces venezuelae* strains typically produce an average of $10^8$ CFUs per mL in chitin cultures, whereas the *vnz_24855* knock-out (KO), *vnz_12765* KO and *vnz_26400* KO reach around $10^6$ CFUs per mL within the same time frame. Complemented strains (CP) for the *vnz_24855* and *12765* chitinases reach comparable CFU's as wild-type cultures. KO's of *vnz_02735* and *16685* maintain similar amounts of spores compared to WT. Bars represent the averages of three biological replicates and their respective error bars show standard deviations. **B)** After 6 days of growth on chitin, wild-type *S. venezuelae* bacteria formed large pellets around the chitin flakes and are sporulating (green pellet indicated by the arrow). In contrast, only few hyphae are visible for the *dasBC* KO. Inserts: pellets from 0.5 mL culture samples. Scale bar: 50 micrometer. The data underlying this figure can be found in S2 Data.

Loss of *dasB* and *dasC* led to a severe growth delay, resulting in no detectable spores after 6 days of growth, indicated by the absence of green spore pigment in the cell pellet. (Fig 3B—bottom left insets). As expected, the growth and development of the *dasBC* mutant was equivalent to wild-type in MYM (Fig H in S1 Text). Microscopic examination of chitin flakes in the cultures (Fig 3B) revealed some residual growth for the *dasBC* deletion mutant. We presume that this residual growth stemmed from the uptake of other products of chitin degradation.

These data demonstrate that *S. venezuelae* possesses an elaborate chitinolytic system where members of the group A and group B chitinases play important roles in generating metabolizable material from insoluble chitin. It further shows that the DasABC complex is important for transporting the products of this catabolic activity into cells for further metabolism.

## The Das transporter prevents resource-sharing

The profound effect of the *dasBC* mutation on the *S. venezuelae* life cycle supported a critical role for this transporter in chitin catabolism. We were intrigued, however, by the ability of the mutant to grow on chitin, albeit at greatly reduced levels, as this suggested that chitinases may still be secreted and functioning to breakdown chitin. If the Das importer is indeed the dominant importer of chitin breakdown products, we imagined that these products might accumulate in culture medium in the absence of uptake capacity. We therefore conducted LC-MS analyses of spent culture supernatant from wild-type and *dasBC* cultures (see materials and methods). After four days of growth, we identified a peak with $m/z = 447.16$ from the *dasBC* culture that was not present in our wild-type samples (Fig 4A). This agrees with the $m/z$ value of the sodium (Na) adduct of chitobiose. Importantly, the spectrum showed the co-occurrence of $m/z = 425.18$ (Fig 4B), corresponding to the $m/z$ value for the M + H ion of chitobiose. Further MS–MS analysis of the 447.16 ion showed fragments consistent with the sodium adduct of the GlcNAc monomer (244.08), as well as the same ion without water (226.07) (Fig 4C); this fragmentation pattern supports the accumulation of chitobiose in the *dasBC* culture. By measuring the area under the curve (AUC), we determined that chitobiose accumulation was at least 20 times greater in the spent medium of the *dasBC* null mutant (Fig 4C—inset) than the wild-type. Thus, even though growth on chitin is greatly compromised, the chitinolytic system in the *S. venezuelae dasBC* deletion mutants is clearly active and is releasing significant amounts of chitobiose.

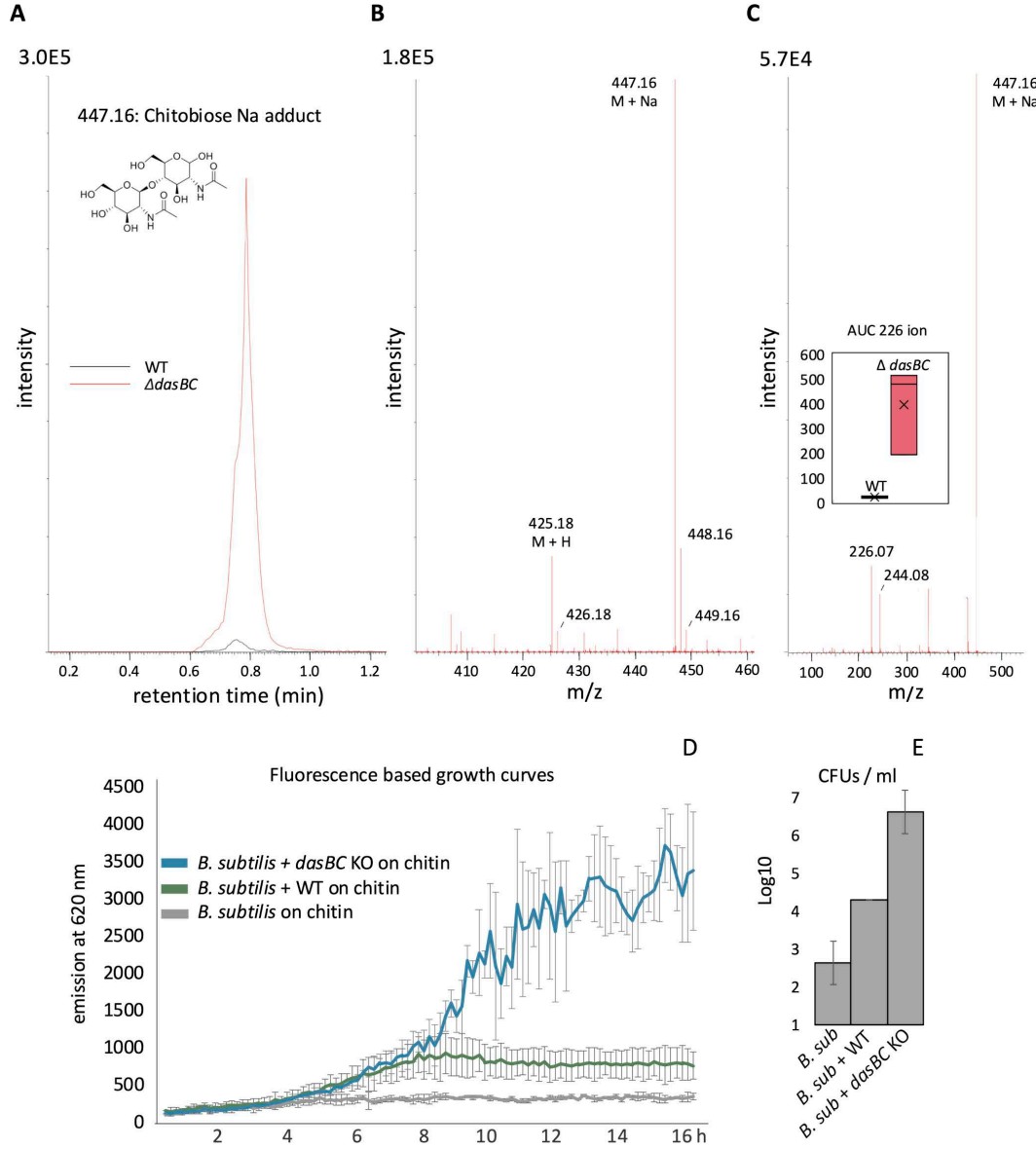

**Fig 4. Chitobiose accumulation in the *dasBC* deletion mutant enhances *Bacillus subtilis* growth in co-culture. A)** Extracted ion chromatogram for chitobiose (N,N″-diacetylglucosamine dimer—structure inserted; M+Na: 447.16). Shown are representative peaks for *S. venezuelae* wild-type (WT) and the deletion mutant (Δ*dasBC*). **B)** Chitobiose spectrum (MS[1]). Besides the M+Na adduct and its isotopes, the M+H ion (425.18) and its isotopes are detected too. **C)** MS/MS spectrum of ion 447.16. Amongst the fragments detected are the ions 244.08 and 226.07, which represent the Na adduct of GlcNAc with and without $H_2O$, respectively. Insert: Area under the curve (AUC) of the 226.07 ion in spent media of wild-type vs. mutant bacteria. **D)** *B. subtilis* growth curves. *B. subtilis* expressing the red fluorophore mCherry does not exhibit growth on chitin medium, as indicated by the absence of fluorescence. When co-cultured with the *Streptomyces venezuelae dasBC* deletion mutant (*dasBC* KO), there is a marked increase in fluorescence output compared to *B. subtilis* grown alone on chitin medium or when co-cultured with wild-type *S. venezuelae* (WT). Datapoints are the averages of three biological replicates and the error bars show standard deviations. **E)** Colony-forming units (CFU's) per milliliter of culture for *B. subtilis*. The number of *B. subtilis* cells is higher in a co-culture with the *S. venezuelae dasBC* deletion mutant compared to *B. subtilis* grown alone or in co-culture with wild-type *S. venezuelae*. Bars represent the averages of three biological replicates and their respective error bars show standard deviations. The data underlying this figure can be found in S2 Data.

We then asked whether *B. subtilis*, which cannot grow in chitin medium (Fig E in S1 Text), could grow when co-cultured with *S. venezuelae*, indicating that the streptomycete shares the resources it liberates. To assess this, we added equal numbers of *B. subtilis* cells and *S. venezuelae* spores to chitin medium. We followed *B. subtilis* growth by measuring the fluorescence of an mCherry transgene ($OD_{600}$ measurements would have been confounded by the presence of *S. venezuelae* and insoluble chitin flakes). When co-cultured with wild-type *S. venezuelae*, little fluorescence was observed, suggesting that *B. subtilis* had little apparent growth. In contrast, co-culture with the *dasBC* deletion mutant supported *B. subtilis* growth, as indicated by an increase in fluorescent signal over time (Fig 4D) and a substantial rise in colony-forming units (Fig 4E). These results suggest that in the absence of a functional DasABC transport complex, chitin breakdown products accumulate extracellularly, making them accessible to other microbes such as *B. subtilis*. *B. subtilis* grew normally during co-culture with both the wild-type *S. venezuelae* and the *dasBC* mutant in MYM medium, showing that co-cultures with *S. venezuelae* are not inherently detrimental to *B. subtilis* (Fig G in S1 Text). *Pseudomonas aeruginosa* and *Rhodococcus jostii* were unable to grow in defined chitin medium even when co-cultured with wild type or *dasBC* mutant of *S. venezuelae.* This implies that, in addition to being unable to use chitin as sole source of carbon, nitrogen and energy, they were not able to use any of the products released during chitin catabolism by *S. venezuelae.*

We did, however, want to address the possibility that the poor *B. subtilis* growth during co-culture with wild-type *S. venezuelae* might be due to antibiotic production by wild-type *Streptomyces* but not the *dasBC* mutant. We therefore grew cultures of the wild-type and *dasBC* mutant on chitin, prepared concentrated spent media extracts, and assessed the growth of *B. subtilis* when exposed to these. Neither the wild-type extract nor the dasBC mutant extract inhibited the growth of *B. subtilis* (Fig I in S1 Text). This observation is consistent with the RNA sequencing data, which indicates that the biosynthetic gene clusters responsible for chloramphenicol production (*vnz_04400 - vnz_04475*) and other secondary metabolites, such as venemycin, watasemycin, and thiazostatin (*vnz_02195 - vnz_02535*), are not transcriptionally activated during the first 24 hours of growth on chitin (S1 Data) [77].

We conclude that the Das transporter serves both to import critical products of extracellular chitin degradation products and, simultaneously, to prevent the sharing of this material, with other microbes. This suggests a novel view of an important ABC transporter's role in nature that we suggest is critical for the evolutionary success of this lifestyle.

## Discussion

This work reveals that *Streptomyces venezuelae* grows remarkably well using insects and insoluble chitinous material as sole source of carbon, nitrogen and energy. A defined chitin medium, composed of chitin, phosphate buffer, and trace elements, supported high metabolic activity in *S. venezuelae*, enabling it to complete its life cycle and degrade over 50% of the chitin present. It was striking that chitin outperformed glucose as a carbon source, given that glucose is typically considered the optimal carbon source for most microbes [78]. The ability to use insoluble chitin as the sole carbon and nitrogen source is conserved across all *Streptomyces* species we tested, including the highly divergent *Streptomyces coelicolor* and *Streptomyces avermitilis*, as well as the environmental isolates WAC288, and WAC303. Chitinolytic features are widespread throughout the *Streptomyces* genus [27,38,52], leading us to suggest that the capacity to use chitin as the sole carbon and nitrogen source is a conserved trait among streptomycetes.

### Chitobiose is a critical bioavailable chitin breakdown product

Growth on chitin in *S. venezuelae* is facilitated by a chitinolytic system consisting of up to 10 chitinases and the DasABC importer for chitobiose. Some chitinase mutations did not confer a growth defect on chitin. Most likely these chitinases have redundant roles with other enzymes or, conceivably, serve different functions, like antifungal activity [79–81]. The chitinase deletion mutants that exhibited a phenotype consistently showed delayed sporulation, indicating that the loss of a single chitinase gene can have tangible effects. Even these effects appear to be mitigated over time under laboratory conditions, likely due to functional redundancy within the system. Importantly, we showed that the DasABC importer is

a crucial part of this metabolic pathway. This suggests that chitobiose is a critical, bioavailable chitin catabolite. Modest growth was, however, possible in the absence of a functional Das ABC transporter, suggesting that at least one other breakdown product can be imported through a currently unknown importer. A reasonable candidate is GlcNAc, which is likely to be released during the chitin degradation process. It has been further established that in *S. coelicolor*, GlcNac is imported via a specialized phosphotransferase system [82]. The gene cluster encoding these genes is also conserved in streptomycetes, including *S. venezuelae* (*vnz_13025 – vnz_13035*). Transcript levels of these genes are notably higher when *S. venezuelae* is grown on chitin compared to glucose (S1 Data).

### The DasABC importer limits chitobiose availability to competitors during chitin catabolism

The DasABC transporter is a known transporter of chitobiose in *S. coelicolor* [51]. BLAST searches confirm its presence in the vast majority of sequenced *Streptomyces* genomes, as well as in *Streptacidiphila* and *Kitasatospora* species. The *S. coelicolor* operon is transcriptionally controlled by the repressor DasR, a pleiotropic regulator of metabolism and antibiotic production [83]. Importantly, DasR relieves repression upon GlcNAc binding. This induces transcription of the *dasABCD* operon, where *dasA* encodes the extracytoplasmic chitobiose-binding protein, while *dasB* and *dasC* encode permeases, and *dasD* encodes an intracellular β-N-acetylglucosaminidase that hydrolyzes the transported chitobiose into GlcNAc subunits.

Importantly, we found that DasABC prevented the accumulation of chitobiose into the environment during extracellular chitin catabolism. These findings highlight a key aspect of *Streptomyces* saprophytic growth: efficient uptake of chitin-derived oligomers limits their availability to other microbes. *B. subtilis* was only able to proliferate on minimal chitin medium when co-cultured with the dasBC deletion mutant, suggesting that the accumulation of extracellular chitobiose in the absence of a functional importer supported its growth. Importantly, while *B. subtilis* encodes chitinase-like enzymes, there is no direct evidence from the available literature that it can import chitobiose or that it encodes a specific chitobiose transporter [84,85]. We therefore ruled out an alternate explanation whereby chitobiose import triggered the biosynthesis of antibiotic compounds exclusively in wild-type S. *venezuelae,* that prevented *B. subtilis* growth. We would note, however, that we have not ruled out the possibility that other compounds released by the *dasBC* mutant during growth on chitin (including GlcNAc oligomers) contribute to the growth of *B. subtilis*.

It is interesting that secondary metabolites did not appear to play a role in the competition for chitin, at least under the conditions we have employed. Work by other investigators [86,87] has demonstrated that some interspecies competition involves the explicit induction of specialized metabolism in streptomycetes. Indeed, *B. subtilis* is highly sensitive to chloramphenicol, one of the specialized metabolites in the *S. venezuelae* armamentarium. In this case, however, the competitive edge was attributed to the activity of an efficient ABC transporter. The density of microbial life in the soil [28,88] and the existence of antibiotic resistance mechanisms in many microbes [89] suggests that robust nutrient sequestration could drive *Streptomyces'* success in their soil environment. Also, the physical structure of soil and its organic particles limits diffusion [90,91]. Therefore, transporter-mediated privatization could reinforce spatial heterogeneity in nutrient availability and promote the formation of microcolonies enriched in saprophytic species like *Streptomyces* [91,92].

### Chitin catabolism on Earth

We show that *Streptomyces* are robust chitin metabolisers, joining the ranks of other established chitin-degrading microbes, like many organisms of the genus *Vibrio* and *Serratia marcescens. S. marcescens* has four experimentally validated chitinolytic enzymes: ChiC, an endo-acting non-processive chitinase, as well as ChiA and ChiB, which are processive chitinases [93]. A fourth enzyme is a lytic polysaccharide monooxygenase that introduces oxidative chain breaks [94]. Our *in silico* analysis suggests that the *S. venezuelae* chitinolytic system encompasses all of these chitin-degrading functions and that the genes responsible for these functionalities are induced upon exposure to chitin (Fig 2C, S1 Data, and Table D in S1 Text). It would be interesting to see whether *S. marcescens* can also utilize insoluble raw chitin as a sole source of carbon and nitrogen.

The *Vibrionaceae* offer the only other well-understood example of chitin metabolism, in this case exclusively found in marine ecosystems [95,96]. Similar to *Streptomyces*, these bacteria can utilize chitin as their sole carbon and nitrogen source [97]. The *Vibrio cholerae* genome also encodes 10 secreted chitinases, and similar to our findings for *S. venezuelae*, some of these enzymes play a significant role in supporting growth on chitin, while others have no considerable effect [98]. In *Vibrio,* secreted chitinases degrade chitin extracellularly into oligosaccharides of varying lengths. The resulting oligosaccharides are imported through a dedicated outer membrane channel called chitoporin. Once in the periplasm, the oligosaccharides are degraded into GlcNAc and chitobiose, which are then transported into the cytoplasm where they enter primary metabolism [97]. Importantly, *Vibrio* forms dense biofilms that prevent the diffusion of these break-down metabolites away from the *Vibrio* cells [99]. Noteworthy, we observed chitin-attached pelleted growth, which can be considered a biofilm in the otherwise fragmented *S. venezuelae*. A further difference; however, is that the *Vibrio* system also includes chitin-responsive chemotaxis proteins that attract cells to chitin-containing substrates [97,100]; whether an analogous system operates in *Streptomyces* remains to be seen. Given the high metabolic value and global abundance of chitin, it would be expected that multiple organisms have devised distinct but effective strategies to it use as a nutrient.

As primary agents in the formation and decomposition of organic matter, soil microorganisms are central to maintaining the global carbon balance [88]. This work demonstrates *Streptomyces'* remarkable capacity for chitin decomposition, providing definitive evidence of this ability and identifying key players involved in this mode of growth. Members of the *Streptomyces* genus appear to achieve this without broadly sharing the products of saprophytic digestion, a trait that has likely contributed to their evolutionary success over the past ~380 million years [101]. Therefore, alongside *Vibrio* and *Serratia, Streptomyces* species have arguably played key roles in the cycling of the earth's nitrogen and carbon, thereby impacting everything from soil nutrient content to climate.

## Supporting information

**S1 Text. Fig A: Correlation of gene-expression between samples plotted into a heatmap.** Gene-expression between samples was measured using Pearson's correlation and plotted into a heatmap. Samples from glucose-grown and chitin-grown conditions (each in triplicate) cluster strongly by condition, indicating high within-condition reproducibility and distinct transcriptional profiles. The data underlying this Figure can be found in S1 Data. **Fig B: Principal Component Analysis RNA seq samples.** First two principal components are plotted. PCA was performed on normalized gene expression values from glucose-grown and chitin-grown samples, grown for 24 h in liquid cultures ($n = 3$ per condition). The first two principal components capture the majority of variance across samples. Chitin-grown replicates cluster tightly together, while glucose-grown samples are slightly more dispersed but still primarily separated along the same components, indicating distinct transcriptional signatures between conditions. The data underlying this Figure can be found in S1 Data. **Fig C: SEM images of *S. venezuelae* growing on grasshopper. A)** Mycelial sheet covering a significant part of the coxa and femur of the midleg. **B)** Closer view of the mycelial sheet edge, showing multiple hyphal layers entirely coating the exoskeleton. **C)** Further magnification of B, depicting an individual hypha navigating between the exoskeleton scales. Scale bar: 25 µM. **Fig D: Mycelial aggregates of various *Streptomyces* species on chitin flakes.** Dense mycelial aggregates on chitin flakes isolated from shrimp were observed after two to four days of growth for **A)** *Streptomyces coelicolor;* **B)** *Streptomyces avermitilis;* **C)** *Streptomyces* sp. WAC288; **D)** *Streptomyces* sp. WAC303. Scale bar: 50 microns. **Fig E. Growth on chitin by *Bacillus subtilis, E. coli, P. aeruginosa,* and *R. jostii*. A)** Growth curves of *B. subtilis* in LB and minimal chitin medium, monitored via mCherry fluorescence. *B. subtilis* shows steady growth in LB, as indicated by increasing fluorescence over time. In contrast, no appreciable increase in fluorescence was observed when *B. subtilis* was cultured in chitin medium. **B)** Colony-forming units (CFUs) of *E. coli*, *P. aeruginosa*, and *R. jostii* after incubation in minimal chitin medium chitin and LB (or MYM for *R. jostii*). Cultures were inoculated with $10^5$ CFUs/mL (indicated by the dotted line) and incubated at 30 °C with shaking for 24 h (*E. coli* and *P. aeruginosa*) or 48 h (*R. jostii*). CFU counts in chitin medium remained stable or declined, whereas in LB/MYM CFUs increased. Data represent the average of three

biological replicates with standard deviations. The data underlying this figure can be found in S2 Data. **Fig F. Phylogeny and expression of *S. venezuelae* chitinases**. Unrooted tree of various known bacterial GH18 chitinases from subfamilies A, B, and C, including *S. venezuelae*. **Fig H: Colony-forming units (CFUs) of wild-type and mutant strains grown in MYM liquid cultures.** Wild-type (WT) strains typically produce an average of $10^9$ CFUs per milliliter in MYM cultures. Mutant strains (KO for knock-out) that are affected during growth on chitin maintain comparable CFUs per microliter on MYM. Bars represent the averages of three biological replicates and the error bars show standard deviations. The data underlying this figure can be found in S2 Data. **Fig G: Co-culture of *B. subtilis*-mCherry with *S. venezuelae* on MYM.** Both wild-type (WT) and the *dasBC* deletion *Streptomyces* strains allow growth of *B. subtilis* on MYM as indicated by the increase of fluorescence signal over time. Datapoints are averages of three biological replicates and the error bars show standard deviations. The data underlying this figure can be found in S2 Data. **Fig I: Growth curves *B. subtilis*-mCherry on LB with *Streptomyces venezuelae* spent medium concentrate.** *B. subtilis* expressing the red fluorophore mCherry is unable to grow on LB medium supplemented with 25 µg/ml chloramphenicol. Growth on LB with either extracts of chitin medium, or concentrated spend medium of *S. venezuelae* wild-type (WT) and the *dasBC* deletion mutant (*dasBC* KO) grown on chitin grown for 16H, resemble growth comparable to when the bacterium is grown on LB alone. Datapoints are averages of three biological replicates and the error bars show standard deviations. The data underlying this figure can be found in S2 Data. **Table A: Strains, plasmids and cosmids used in this study. Table B: Uniprot entries *Streptomyces venezuelae* chitinases. Table C: primers used in this study. Table D: List of 40 genes that are most significantly upregulated during growth on chitin. Table E: Number of hits in comparative BLASTP analysis of *S. venezuelae* recombinase RecA, and Vnz_04420 (of the chloramphenicol biosynthetic gene cluster), and chitinases (Chi) in *Streptomycetaceae*.** * Homologs were identified by BLASTp against the NCBI protein database using full-length protein sequences of all identified *S. venezuelae* chitinases, recombinase Vnz_26845, and Vnz_04420. Inclusion criteria: query coverage 80%–100%, sequence identity 50%–100%, *E*-value ≤ 1E−05, and a maximum of 5,000 target sequences retrieved.
(DOCX)

**S1 Data. Differential expression values, counts per million, and significance of genes for growth on glucose vs. growth on chitin.**
(ZIP)

**S2 Data. Raw data supporting** Figs 1B, 1F, 1G, 2A, 2B, 3A, 4A, 4B, 4C, 4D; **Figs A, B, E(panel A), E(panel B), G, H and I in S1 Text**.
(XLSX)

**S3 Data. Treefile for Fig 2C.**
(NWK)

**S4 Data. Treefile for Fig F in S1 Text.**
(NWK)

## Acknowledgments

We would like to thank Prof. D.Z. Rudner for kindly providing us with *Bacillus subtilis* BDR2662. We also thank Prof. Rosie Redfield for critical appraisal of an early draft of this work.

## Author contributions

**Conceptualization:** Anne van der Meij, Justin R. Nodwell.

**Data curation:** Anne van der Meij, Dustin J. Sokolowski.

---

   

**Formal analysis:** Anne van der Meij, Dustin J. Sokolowski, Justin R. Nodwell.

**Funding acquisition:** Anne van der Meij, Justin R. Nodwell.

**Investigation:** Anne van der Meij, Hannah Tyrrell, Justin R. Nodwell.

**Methodology:** Anne van der Meij, Hannah Tyrrell, Evan M. F. Shepherdson, Marie A. Elliot, Justin R. Nodwell.

**Project administration:** Anne van der Meij, Justin R. Nodwell.

**Resources:** Marie A. Elliot, Justin R. Nodwell.

**Supervision:** Anne van der Meij, Justin R. Nodwell.

**Validation:** Anne van der Meij.

**Visualization:** Anne van der Meij, Justin R. Nodwell.

**Writing – original draft:** Anne van der Meij, Justin R. Nodwell.

**Writing – review & editing:** Anne van der Meij, Marie A. Elliot, Justin R. Nodwell.

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
