## [Editor Report · Decision Letter 0]

21 Mar 2025

Dear Dr Nodwell,

Thank you for submitting your manuscript entitled "A metabolic pathway for competitive saprophytic metabolism of chitin by terrestrial bacteria" for consideration as a Research Article by PLOS Biology.

Your manuscript has now been evaluated by the PLOS Biology editorial staff, as well as by an academic editor with relevant expertise, and I'm writing to let you know that we would like to send your submission out for external peer review.

IMPORTANT: We think that your manuscript would be better considered as a Discovery Report (https://journals.plos.org/plosbiology/s/what-we-publish#loc-discovery-report). Your paper is already concise, but please can you reduce the number of Figures from 6 to 4, either by combining the current Figs into multi-panel Figs, or by moving less crucial material to the supplement. Please also select "Discovery Report" as the article type when you upload the modified manuscript and your additional metadata (see next paragraph).

Once your full submission is complete, your paper will undergo a series of checks in preparation for peer review. After your manuscript has passed the checks it will be sent out for review. To provide the metadata for your submission, please Login to Editorial Manager (https://www.editorialmanager.com/pbiology) within two working days, i.e. by Mar 25 2025 11:59PM.

Kind regards,

Roli Roberts

Roland Roberts, PhD

Senior Editor

PLOS Biology

rroberts@plos.org

---

## [Decision Letter · Decision Letter 1]

14 May 2025

Dear Justin,

Thank you for your patience while your manuscript "A metabolic pathway for competitive saprophytic metabolism of chitin by terrestrial bacteria" went through peer-review at PLOS Biology. Your manuscript has now been evaluated by the PLOS Biology editors, an Academic Editor with relevant expertise, and by four independent reviewers. Please accept my apologies for the additional delay incurred while we discussed the reviews with the Academic Editor.

Reviewer #1 is positive and has largely textual requests, but he suggests co-culturing with an additional soil bacterium (their point 2). Reviewer #2 is also positive, with mostly textual and presentational requests, but several points involve additional data and/or analyses. Reviewer #3 only has textual requests. Likewise reviewer #4, though there are some optional experimental suggestions here.

In light of the reviews, which you will find at the end of this email, we are pleased to offer you the opportunity to address the comments from the reviewers in a revision that we anticipate should not take you very long. We will then assess your revised manuscript and your response to the reviewers' comments with our Academic Editor aiming to avoid further rounds of peer-review, although we might need to consult with the reviewers, depending on the nature of the revisions.

**IMPORTANT - SUBMITTING YOUR REVISION**

*Resubmission Checklist*

*Published Peer Review*

*PLOS Data Policy*

*Blot and Gel Data Policy*

Sincerely,

Roli

Roland Roberts, PhD

Senior Editor

PLOS Biology

rroberts@plos.org

REVIEWERS' COMMENTS:

Reviewer #1:

[identifies himself as Glen D'Souza]

This manuscript presents a well-executed and compelling investigation into the chitinolytic metabolism of Streptomyces venezuelae. The authors demonstrate that S. venezuelae can use raw, insoluble chitin as a sole carbon and nitrogen source, outperforming glucose in both metabolic activity and sporulation. They identify a suite of chitinases and establish the central role of the DasABC transporter in chitobiose uptake—both for metabolic competence and for limiting nutrient access to Bacillus subtilis in co-culture. The integration of transcriptomics, targeted mutagenesis, LC-MS, and microbial competition experiments strengthens the manuscript substantially. Overall, this study makes a novel and significant contribution to microbial ecology, physiology, and our understanding of nutrient competition in soil ecosystems.

I have some points where the manuscript can be improved further.

1. The story has broad biological and ecological relevance, but the title is narrowly focused. I encourage the authors to consider a more general title, such as: Ecological Specialization in Chitin Catabolism by Streptomyces Supports Competitive Growth and Nutrient Sequestration, or A Conserved Strategy for Competitive Chitin Metabolism in Soil Bacteria

2. The authors convincingly show that DasABC restricts chitobiose access to B. subtilis, but testing an additional soil microbe (e.g., Pseudomonas fluorescens) in co-culture would substantially broaden the ecological implications and reinforce the generality of nutrient privatization.

3. Some chitinases showed no growth phenotype when deleted. A short discussion on whether this reflects functional redundancy, context-dependent activity (e.g., biofilm-associated, substrate-dependent), or differential regulation would be valuable.

4. The authors should clarify whether the B. subtilis strain used encodes known chitobiose transporters. This would help contextualize its ability to exploit the extracellular chitobiose in dasBC mutant co-cultures.

5. The RNA-seq data analysis is informative, but the link between differential gene expression and the observed physiological outcomes could be drawn more clearly. For instance, discussion of regulatory patterns, carbon utilization pathways, or broader stress responses would enrich the interpretation.

6. The manuscript infers that DasABC is specific for chitobiose. However, given the variety of oligosaccharides released during chitin degradation, please clarify whether the transporter might import longer oligomers, and to what extent its selectivity has been tested or inferred from the data.

7. The study highlights the role of ABC transporters in nutrient privatization. A short speculative note on how such transporters may contribute to spatial structuring or stability of microbial communities would place the findings in a broader ecological context.

8. The manuscript notes that insect exoskeletons contain proteins and carbohydrates in addition to chitin. It would help to briefly comment on whether these components are likely to influence growth outcomes or if their contribution is negligible compared to chitin.

Reviewer #2:

Saprophytic processes play a key role in global carbon and nitrogen cycles. In this manuscript, the authors studied the chitinolytic capacities of Streptomyces venezuelae. Chitin utilization by Streptomyces species was previously shown, but typically focused on colloidal chitin and frequently in the presence of other C- or N-sources.

In this study, the authors demonstrate growth of S. venezuelae and further Streptomyces species on raw chitin (grasshoppers) and showed an even higher „metabolic activity" on chitin than compared to MYM medium. Transcriptome analysis revelead the upregulation of several chitinases and mutational analysis confirmed the relevance of several chitinases for chitin consumption. They furthermore highlight the relevance of the DasABC transporter, which was previously described as a chitobiose importer in S. coelicolor. Intriguingly, they show that the activity of this importer enables S. venezuelae to grow competitively on chitin in the presence of bacterial competitors such as Bacillus subtilis. A deletion of dasABC led to the accumulation of chitobiose in the culture supernatant and supported growth of B. subtilis.

Overall, this study offers significant insights into the molecular mechanisms underlying saprophytic lifestyles and is broadly relevant to our understanding of microbial ecology and nutrient cycling. I have only minor comments for revision of this well-written manuscript.

1) L. 60-68: the authors introduce the definition of „cheaters" and „exploiters", but the difference is not really clear to me. Also in siderophore-mediated interactions both terms exist. While a cheater is a mutant derived from a producer which lost the capability of producing the siderophore, but can still utilize it, an exploiter is specialized on the utilization of xenosiderophores. However, since both terms are not really used in the following manuscript, it only needs a brief introduction into nutrient competition and privatization strategies.

2) L. 341-347; Fig. S5: Can B. subtilis grow on the minimal (chitin) medium in the presence of glucose? To exclude lack of relevant trace elements.

3) L. 366-372, please indicate whether cultures were grown on plates or in liquid cultures for transcriptomics.

4) L.402-413, the authors discuss the different chitinases encoded by S. venezuelae. I do find this information quite critical for the manuscript and would suggest to include Figure S6 in the main text.

5) Overall, Figures can be produced with a bit more care (e.g. harmonization of font sizes and styles, etc.)

6) Figure 1: The finding that S. venezuelae shows better growth on chitin than on glucose is quite intruiging. While I do understand that simple quantification via OD is not possible, I would suggest to include CFUs time course measurements to support the conclusions based on the resazurin measurement (Fig 1G)

7) Figure 2: please include the information to all COG categories shown in the Figure.

8) Figure 3: Please include significance tests for Fig 3A. The Streptomyces CFUs represent the number of mycelial pellets. Are those similar in size for the different mutants?

9) Figure 4: Please include growth of B. subtilis on chitibiose to support the „resource-sharing" experiments, although already shown indirectly with the dasBC KO.

10) Figure 4D: please include a proper x axis for the time! Y axis: please indicate the emission wavelength. I would further suggest to include B. subtilis (Fig. S5) here for direct comparison.

Supplemental material:

- Please include informative Figure legends for all figures, in particular Figure S1 and S2 further experimental details is needed to avoid hopping there and back between SI and main text.

- Figure S4: it is not really clear what is seen here. It is not „growth", but rather mycelial structures. Please include further information. It would be intriguing to see whether the resazurin assay or CFU counting assays would show the same trend for other Streptomyces species.

- Figure S5, legend: "A similar increase…" ?

- Figure S9: „spent medium"; Please indicate the time of harvesting the different spent media.

Reviewer #3:

The paper nicely demonstrates the ability of streptomycetes to utilize and grow on insoluble chitin polymers. Gene expression patterns associated with growth on chitin are reported and the chitinase genes in the model organism S. venezuelae are identified. Not unexpectedly, three of the most highly expressed chitinase genes are needed for good growth on raw chitin. Further the transporter DasABC is shown to be involved in uptake of degradation products from chitin, including chitobiose. When DasABC is inactivated, chitobiose (and perhaps other soluble products of chitin degradation) are accumulated extracellularly and can be utilized by other bacteria, as demonstrated here by allowing growth of B. subtilis.

Technically, the work and the documentation of it are great quality, and the conclusions are founded on clear and conclusive data. Regarding the overall question, it is, like the authors make clear in the introduction, no novelty that streptomycetes often like to grow on chitin (at least the colloidal from) and have chitinases. However, this paper adds a comprehensive investigation of the growth on raw, insoluble chitin, and a test of the roles of specific chitinases and the DasABC uptake system. It provides also insight into the significance of uptake systems in the life-style of saprophytic microbes in general and their ability to compete in the environment.

I have only a few minor comments and suggestions.

1. The effects of mutations on growth in chitin medium were estimated by assaying numbers of cfu in cultures after a certain time. This method has its limitations, particularly in cultures with large degree of clumping, as clearly is the case in the medium based on chitin flakes. Another concern is that sporulation will have a very large effect on the numbers of cfu in cultures since it would release single-cell spores from the clumps. Thus, the results of the growth assays in Fig. 3A should be further clarified. Are the large differences in cfu in some chitinase mutants when compared to WT due to differences in ability to sporulate under these conditions, or do they reflect true differences in growth and production of biomass? The authors likely have more insights or perhaps data to clarify the results (like is done for the dasBC mutants in Fig. 3B).

2. Given that there are ten chitinases in this organism, is it not surprising that deletions of single chitinase genes have such large effects? One could have expected to see substantial redundancy. Or is it possible that several of the chitinases have specific roles and that many of them are required? This could be worth discussing.

3. A list of strains and plasmids is missing. Would be valuable to add to the Supplementary material.

Minor

* In legends to Fig. 1B and 1F, please mention how long time of incubation before the measurements.

* Line 336: Why reference to Fig. 4A here? Typo?

* Lines 118 and 447: This is not the right reference for MYM medium.

* Legend to Fig. S4: Good to say what chitin this was or what type of growth conditions.

* Similarly, it would help the reader if the criteria for inclusion in Table S5 were mentioned in the Table, for example as a footnote.

* Slight rephrasing could be done in the two final sentences (lines 636-640). Streptomyces is a group of organisms, and it becomes a bit strange (and maybe not the best style) to talk about them as a single entity. ("Streptomyces achieves…"; "its evolutionary success…"; "Streptomyces plays key roles").

Reviewer #4:

Summary: van der Meij et. al. demonstrate that the model streptomycete S. venezuelae can effectively colonize insect caracasses, metabolize chitin, and compete with potential exploiters for liberated resources from extracellular chitin degradation. Importantly, they identify multiple extracellular chitin degrading enzymes and demonstrate that effective competition for liberated catabolites results from efficient uptake via the DasABC transporter. In all, the authors offer a compelling account of chitin utilization by streptomycetes, demonstrating that these organisms may play in important role in terrestrial chitin turnover. My reaction to this work is overwhelmingly positive; I have only one major concern (which is not even that major), and a few minor concerns. Most of my minor concerns revolve around term usage, which I think is a little loose throughout the manuscript. Beyond this, the paper is very well written and I quite enjoyed reading it.

Major concerns: My only major concern is with the title, which starts with the phrase "A metabolic pathway". This sets the expectation the paper will offer a new metabolic pathway for chitin metabolism. There are no pathways in this paper. I might suggest that the title could simply be: "Competitive saprophytic metabolism of chitin by terrestrial bacteria"

Minor concerns:

L12: Why is the N and C in chitin considered fixed? The organisms with exoskeletons are fixing neither. In this sense, chitin is like any other C/N source.

L14: Chitin is indeed found in exoskeletons, but what about fungal biomass? My feeling is that chitin in fungal biomass probably far exceeds the amount found in arthropod exoskeletons.

L60: "cheaters". I am taking issue with 'cheaters' here. I suggest that the word cheater implies that that they were once a cooperator that is now defecting, either through mutation or behavior. In the case of siderophore piracy, there is no prior expectation of cooperation. Almost all bacteria have uptake systems for siderophores that they cannot make. Are they all cheaters? I do not believe so. I think that the word 'pirate' or 'exploiter' is more appropriate here. This restrictive definition of cheating can be seen in the review:

Michael E. Hibbing, Clay Fuqua, Matthew R. Parsek & S. Brook Peterson. Bacterial competition: surviving and thriving in the microbial jungle. Nature Reviews Microbiology volume 8, pages15-25 (2010).

The authors cite a review (29), which does use 'cheating' to encompass siderophore piracy. However, I would suggest that this an example of 'semantic bleaching', which is the gradual weakening or reduction of a word's meaning over time, wherein the word becomes less specific and more general.

L63: "cheater" again. I don't think 'cheaters' here is correct since there is no expectation or history of prior cooperation. The authors correctly label them 'exploiters' in L64. I think the authors should roll with that.

L73: 'fixed C and N': again, aren't all biological polymers made from fixed C and N? I think none of the organisms that produce chitin fix C or N directly.

L336: Fig 1E: Maybe consider adding an arrow that points at the green spore pellet? To a naive reader, it's not that obvious. Also, since there is no comparison here, it's kind of difficult to say what this means.

L363: In what sense is chitin chemically inert? Noble gasses are inert since they do not participate in chemical reactions. I do not think the same is true of chitin. Certainly, as the authors note, chitin is insoluble and challenging for most microbes to metabolize.

L479: Does this mean that there were no CFUs for the das mutants, as would be assessed in 4A? Do the CFU assays only track spore count, or vegetative growth as well? If so, it should be clarified that the CFU assays only assess spore production. Seems like we are limited to the relatively subjective microscopic inspection for the most important set of mutants here. Not suggesting more experimental work should be done, just asking for clarification.

L558: Did you try the das mutants for growth on the grasshoppers? I imagine they would a have a very large defect in carcass colonization. Again, not suggesting more experimental work, but I'd love to see it!

L584: "efficient uptake of chitin-derived oligomers limits their availability to other microbes." I would suggest that rapid uptake of liberated resources would be subject to selection. Ie. if I am going to spend resources on extracellular degradation, my uptake systems NEED to be very fast to compete against exploiters. Otherwise, this strategy will not work since I would simply lose out to said exploiters.

L593: No observed role for specialized metabolism. This is a bit tricky. My feeling is that 24 hours of growth is probably too short to see the effect of antibiotics, since they would typically be made later. But, then again, the authors did see robust sporulation. My point is that if the experiment went on longer, an effect of specialized metabolism might have been seen. The connection between GlcNAc metabolism and specialized metabolism alluded to by the authors in line 575 is a further hint that a connection to overall chitin metabolism might be observed in longer trials.

L605: "Vibrio's". I'm not sure if this needs an apostrophe. I think the authors might mean 'many organisms of the genus Vibrio'.

L640: "impacting everything from soil nutrient content to climate." I think this line would be more impactful if the authors provided some idea about the amount of C and N flux through chitin, if those estimates exist.

---

## [Editor Report · Decision Letter 2]

16 Jun 2025

Dear Justin,

Thank you for your patience while we considered your revised manuscript "Competitive saprophytic metabolism of chitin by terrestrial bacteria" for publication as a Discovery Report at PLOS Biology. This revised version of your manuscript has been evaluated by the PLOS Biology editors, and the Academic Editor.

Based on our Academic Editor's assessment of your revision, we are likely to accept this manuscript for publication, provided you satisfactorily address the remaining point raised by the Academic Editor, and the following data and other policy-related requests.

a) Please change your Title to something more declarative, with an active verb. Simply flipping it around might work, so we suggest something like, "A terrestrial bacterium degrades chitin through competitive saprophytic metabolism."

b) The Academic Editor wants you to "add the finding to the manuscript that no other environmental isolates were found that could make use of the by-products of chitin" - specifically, more of what you say in your previous response to reviewer #1's point 2. Essentially, as I understand it, you already now say that bacteria X, Y and Z didn't grow on chitin medium (as shown in Fig S5B), but the Academic Editor wants you to also say that bacteria X, Y and Z didn't grow on chitin medium in the presence of S. venezueliae (indeed, S. venezueliae overgrew them), even the dasBC mutant. I know that this is essentially a negative result, but the Academic Eitor saw it as nevertheless being of value.

c) Please address my Data Policy requests below; specifically, we need you to supply the numerical values underlying Figs 1BFG, 2AB, 3A, 4ABCD, S1, S2, S5AB, S6 (treefile), S7, S8, S9, either as a supplementary data file or as a permanent DOI’d deposition.

d) Please cite the location of the data clearly in all relevant main and supplementary Figure legends, e.g. “The data underlying this Figure can be found in S1 Data” or “The data underlying this Figure can be found in https://zenodo.org/records/XXXXXXXX

e) Please make any custom code available, either as a supplementary file or as part of your data deposition.

We expect to receive your revised manuscript within two weeks.

*Published Peer Review History*

*Press*

Sincerely,

Roli

Roland Roberts, PhD

Senior Editor

rroberts@plos.org

PLOS Biology

DATA POLICY:

Regardless of the method selected, please ensure that you provide the individual numerical values that underlie the summary data displayed in the following figure panels as they are essential for readers to assess your analysis and to reproduce it: Figs 1BFG, 2AB, 3A, 4ABCD, S1, S2, S5AB, S6 (treefile), S7, S8, S9.

APPROPRIATE DATA PRESENTATION?

- Please ensure that you are using best practice for statistical reporting and data presentation. These are our guidelines https://journals.plos.org/plosbiology/s/best-practices-in-research-reporting#loc-statistical-reporting and a useful resource on data presentation https://journals.plos.org/plosbiology/article?id=10.1371/journal.pbio.1002128

- If you are reporting experiments where n ≤ 5, please plot each individual data point.

CODE POLICY

Per journal policy, if you have generated any custom code during the course of this investigation, please make it available without restrictions. Please ensure that the code is sufficiently well documented and reusable, and that your Data Statement in the Editorial Manager submission system accurately describes where your code can be found. [IF

DATA NOT SHOWN?

---

## [Editor Report · Decision Letter 3]

3 Jul 2025

Dear Justin,

Thank you for the submission of your revised Discovery Report "Streptomyces venezuelae uses secreted chitinases and a designated ABC transporter to support the competitive saprophytic catabolism of chitin" for publication in PLOS Biology. On behalf of my colleagues and the Academic Editor, Sara Mitri, I'm pleased to say that we can in principle accept your manuscript for publication, provided you address any remaining formatting and reporting issues. These will be detailed in an email you should receive within 2-3 business days from our colleagues in the journal operations team; no action is required from you until then. Please note that we will not be able to formally accept your manuscript and schedule it for publication until you have completed any requested changes.

IMPORTANT: I uploaded the supplementary file that you sent me (AvdM_table S7_Jun20.xlsx) and checked it. It's mostly OK, but I've asked my colleagues to include the following request alongside their own:

"Many thanks for including much of the underlying data. However, I note that for Figs 1BF, 2B, 3A, 4E, S5B and S7 you only provide means and SDs; please supply the values used to calculate these. Please also cite the two treefiles (as 'S1 Data' or similar) in the relevant Figure legends."

Sincerely, 

Roli

Senior Editor

PLOS Biology

rroberts@plos.org